# Human Health during Space Travel: State-of-the-Art Review

**DOI:** 10.3390/cells12010040

**Published:** 2022-12-22

**Authors:** Chayakrit Krittanawong, Nitin Kumar Singh, Richard A. Scheuring, Emmanuel Urquieta, Eric M. Bershad, Timothy R. Macaulay, Scott Kaplin, Carly Dunn, Stephen F. Kry, Thais Russomano, Marc Shepanek, Raymond P. Stowe, Andrew W. Kirkpatrick, Timothy J. Broderick, Jean D. Sibonga, Andrew G. Lee, Brian E. Crucian

**Affiliations:** 1Department of Medicine and Center for Space Medicine, Section of Cardiology, Baylor College of Medicine, Houston, TX 77030, USA; 2Translational Research Institute for Space Health, Houston, TX 77030, USA; 3Department of Cardiovascular Diseases, New York University School of Medicine, New York, NY 10016, USA; 4Biotechnology and Planetary Protection Group, Jet Propulsion Laboratory, California Institute of Technology, Pasadena, CA 91109, USA; 5Flight Medicine, NASA Johnson Space Center, Houston, TX 77058, USA; 6Department of Emergency Medicine and Center for Space Medicine, Baylor College of Medicine, Houston, TX 77030, USA; 7Department of Neurology, Center for Space Medicine, Baylor College of Medicine, Houston, TX 77030, USA; 8KBR, Houston, TX 77002, USA; 9Department of Dermatology, Baylor College of Medicine, Houston, TX 77030, USA; 10Department of Radiation Physics, University of Texas MD Anderson Cancer Center, Houston, TX 77030, USA; 11InnovaSpace, London SE28 0LZ, UK; 12Office of the Chief Health and Medical Officer, NASA, Washington, DC 20546, USA; 13Microgen Laboratories, La Marque, TX 77568, USA; 14Department of Surgery and Critical Care Medicine, University of Calgary, Calgary, AB T2N 1N4, Canada; 15Florida Institute for Human and Machine Cognition, Pensacola, FL 32502, USA; 16Division of Biomedical Research and Environmental Sciences, NASA Lyndon B. Johnson Space Center, Houston, TX 77058, USA; 17Department of Ophthalmology, University of Texas Medical Branch School of Medicine, Galveston, TX 77555, USA; 18Department of Ophthalmology, Blanton Eye Institute, Houston Methodist Hospital, Houston, TX 77030, USA; 19Department of Ophthalmology, University of Texas MD Anderson Cancer Center, Houston, TX 77030, USA; 20Department of Ophthalmology, Texas A and M College of Medicine, College Station, TX 77807, USA; 21Department of Ophthalmology, University of Iowa Hospitals and Clinics, Iowa City, IA 52242, USA; 22Departments of Ophthalmology, Neurology, and Neurosurgery, Weill Cornell Medicine, New York, NY 10021, USA; 23National Aeronautics and Space Administration (NASA) Johnson Space Center, Human Health and Performance Directorate, Houston, TX 77058, USA

**Keywords:** human health, space travel, space mission, space exploration, space radiation, microgravity

## Abstract

The field of human space travel is in the midst of a dramatic revolution. Upcoming missions are looking to push the boundaries of space travel, with plans to travel for longer distances and durations than ever before. Both the National Aeronautics and Space Administration (NASA) and several commercial space companies (e.g., Blue Origin, SpaceX, Virgin Galactic) have already started the process of preparing for long-distance, long-duration space exploration and currently plan to explore inner solar planets (e.g., Mars) by the 2030s. With the emergence of space tourism, space travel has materialized as a potential new, exciting frontier of business, hospitality, medicine, and technology in the coming years. However, current evidence regarding human health in space is very limited, particularly pertaining to short-term and long-term space travel. This review synthesizes developments across the continuum of space health including prior studies and unpublished data from NASA related to each individual organ system, and medical screening prior to space travel. We categorized the extraterrestrial environment into exogenous (e.g., space radiation and microgravity) and endogenous processes (e.g., alteration of humans’ natural circadian rhythm and mental health due to confinement, isolation, immobilization, and lack of social interaction) and their various effects on human health. The aim of this review is to explore the potential health challenges associated with space travel and how they may be overcome in order to enable new paradigms for space health, as well as the use of emerging Artificial Intelligence based (AI) technology to propel future space health research.

## 1. Introduction

Until now space missions have generally been either of short distance (Low Earth Orbit—LEO) or short duration (Apollo Lunar Missions). However, upcoming missions are looking to push the boundaries of space travel, with plans to travel for longer distances and durations than ever before. Both the National Aeronautics and Space Administration (NASA) and several commercial space companies (e.g., Blue Origin, SpaceX) have already started the process of preparing for long distance, long-duration space exploration and currently plan to explore inner solar planets (e.g., Mars) by the 2030s [1].

Within the extraterrestrial environment, a multitude of exogenous and endogenous processes could potentially impact human health in several ways. Examples of exogenous processes include exposure to space radiation and microgravity while in orbit. Space radiation poses a risk to human health via a number of potential mechanisms (e.g., alterations of gut microbiome biosynthesis, accelerated atherosclerosis, bone remodeling, and hematopoietic effects) and prolonged microgravity exposure presents additional potential health risks (e.g., viral reactivation, space motion sickness, muscle/bone atrophy, and orthostatic intolerance) [2,3,4,5,6,7]. Examples of endogenous processes potentially impacted by space travel include alteration of humans’ natural circadian rhythm (e.g., sleep disturbances) and mental health disturbances (e.g., depression, anxiety) due to confinement, isolation, immobilization, and lack of social interaction [8,9,10]. Finally, the risk of unknown exposures, such as yet undiscovered pathogens, remain persistent threats to consider. Thus, prior to the emergence of long distance, long duration space travel it is critical to anticipate the impact of these varied environmental factors and identify potential mitigating strategies. Here, we review the available medical literature on human experiments conducted during space travel and summarize our current knowledge on the effects of living in space for both short and long durations of time. We also discuss the potential countermeasures currently employed during interstellar travel, as well as future directions for medical research in space.

### 1.1. Medical Screening and Certification Prior to Space Travel

When considering preflight medical screening and certification, the requirements and recommendations vary based on the duration of space travel. Suborbital spaceflight, part of the new era of space travel, has participants launching to the edge of space (defined as the Karman line, 100 Km above mean sea level) for brief 3–5 min microgravity exposures. Orbital spaceflight, defined as microgravity exposure for up to 30 days, involves healthy individuals with preflight medical screening. In addition to a physical examination and metabolic screening, preflight medical screening assessing aerobic capacity (VO_2max_), and muscle strength and function may be sufficient to ensure proper conditioning prior to mission launch [11,12,13,14]. Age-appropriate health screening tests (e.g., colonoscopy, serum prostate specific antigen in men, and mammography in women) are generally recommended for astronauts in the same fashion as their counterparts on Earth. In individuals with cardiovascular risk factors or with specific medical conditions, additional screening may be required [15]. The goal of these preflight screening measures is to ensure that medical conditions that may result in sudden incapacitation are identified and either disqualified or treated before the mission begins. In addition to the medical screening described above, short-duration space travelers are also required to undergo acceleration training, hypobaric and hypoxia exposure training, and hypercapnia awareness procedures as part of the preflight training phase.

In preparation for long-duration space travel, astronauts generally undergo a general physical examination, as well as imaging and laboratory studies at the time of initial selection. These screening tests would then be repeated annually, as well as upon assignment to an International Space Station (ISS) mission. ISS crew members are medically certified for long-duration spaceflight missions through individual agency medical boards (e.g., NASA Aerospace Medical Board) and international medical review boards (e.g., Multilateral Space Medicine Board) [16,17]. In order for an individual to become certified for long-duration space travel, an individual must be at the lowest possible risk for the occurrence of medical events during the preflight, infight, and postflight periods. Following spaceflight, it is recommended that returning astronauts undergo occupational surveillance for the remainder of their lifetime for the detection of health issues related to space travel (e.g., NASA’s Lifetime Surveillance of Astronaut Health program) [18]. Table 1 summarizes the preflight, inflight, and postflight screening recommendations for each organ system. Further research utilizing data from either long-term space missions or simulated environments is required in order to develop an adequate preflight scoring system capable of predicting inflight and postflight health outcomes in space travelers based on various risk factors.

Below we discuss potential Space Hazards for each organ system along with possible countermeasures (Table 2). Table 3 lists prospective opportunities for artificial intelligence (AI) implementation.

### 1.2. Effects on the Cardiovascular System

During short-duration spaceflight, microgravity alters cardiovascular physiology by reducing circulatory blood volume, diastolic blood pressure, left ventricular mass, and cardiac contractility [42,123]. Several studies have demonstrated that peak exercise performance is reduced both inflight and immediately after short-duration spaceflight due primarily to a reduction in maximal cardiac output and O_2_ delivery [124,125]. Prolonged exposure to microgravity does cause unloading of the cardiovascular system (e.g., removal of expected loading effects from Earth’s gravity when upright during the day), resulting in cardiac atrophy. These changes may be an example of adaptive physiologic changes (“physiologic atrophy”) that returns to baseline after returning from spaceflight. This process may be similar to the adaptive physiologic changes to the cardiovascular system seen during athletic training (“physiologic hypertrophy”). Thus far, there is no evidence that the observed short-term cardiac atrophy could permanently impair systolic function. However, this physiologic adaptation to microgravity in space could lead to orthostatic hypotension/intolerance upon returning to Earth’s gravity due to changes in the comparative position of peripheral resistance and sympathetic nerve activity [41,126,127]. Figure 1 demonstrates potential effects of the space environment on each organ system.

Another potential effect of microgravity exposure is that an alteration of hydrostatic forces in the vertical gravitational (Gz) axis could lead to the formation of internal jugular vein thromboses [28,29]. Anticoagulation would not be an ideal choice for prevention as astronauts have an increased risk of suffering traumatic injury during spaceflight, thus potentially inflating the risk of developing an intracerebral hemorrhage or subdural hematoma. In addition, if a traumatic accident were to occur during spaceflight, the previously discussed cardiovascular adaptations could impair the body’s ability to tolerate blood loss and shock [45,46,47].

During long-duration spaceflight, one recent study demonstrated that astronauts did not experience orthostatic hypotension/intolerance during routine activities or after landing following 6 months in space [128]. It is worth noting that all of these astronauts performed aggressive exercise countermeasures while in flight [128]. Another study of healthy astronauts after 6 months of space travel showed that the space environment caused transient changes in left atrial structure/electrophysiology, increasing the risk of developing atrial fibrillation (AF) [129]. However, there was no definitive evidence of increased incidence of supraventricular arrhythmias and no identified episodes of AF [129]. Evaluation with echocardiography or cardiac MRI may be considered following long-duration spaceflight in certain cases.

Prior human studies with supplemental data obtained from animal studies, have shown that healthy individuals with prolonged exposure to ionizing radiation may be at increased risk for the development of accelerated atherosclerosis secondary to radiation-induced endothelial damage and a subsequent pro-inflammatory response [3,4,57,58,59,60,123]. One study utilizing human 3D micro-vessel models showed that ionizing radiation inhibits angiogenesis via mechanisms dependent on the linear energy transfer (LET) of charged particles [130], which could eventually lead to cardiac dysfunction [131,132]. In fact, specific characteristics of the radiation encountered in space may be an important factor to understanding its effects. For example, studies of pediatric patients undergoing radiotherapy have shown an increase in cardiac-related morbidity/mortality due to radiation exposure, but not until radiation doses exceeded 10 Gy [133]. At lower dose levels the risk is less clear: while a study of atomic bomb survivors with more than 50 years of followup demonstrated elevated cardiovascular risks at doses < 2 Gy [134]. A recent randomized clinical trial with a 20-year follow-up showed no increase in cardiac mortality in irradiated breast cancer patients with a median dose of 3.0 Gy (1.1–8.1 Gy) [135]. The uncertainty in cardiovascular effects of ionizing radiation, are accentuated in a space environment as the type and quality of radiation likely play an important role as well.

Further research is required to understand the radiation dosage, duration, and quality necessary for cardiovascular effects to manifest, as well as develop preventive strategies for AF and internal jugular vein thrombosis during space travel.

### 1.3. Effects on the Gastrointestinal System

During short-duration spaceflight, the presence of gastrointestinal symptoms (e.g., diarrhea, vomiting, and inflammation of the gastrointestinal tract) are common due to microgravity exposure [35,136,137]. Still unknown however is whether acute, surgical conditions such as cholecystitis and appendicitis occur more frequently due to microgravity-induced stone formation or alterations in human physiology/anatomy, and immunosuppression [40]. Controlling for traditional risk factors associated with the development of these conditions (e.g., adequate hydration, maintenance of a normal BMI, dietary fat avoidance, etc.) may help mitigate the risk.

During long-duration spaceflight, it is possible that prolonged radiation exposure could lead to radiation-induced gastrointestinal cancer. Gamma radiation exposure is a known risk factor for colorectal cancer via an absence of DNA methylation [138]. NASA has recently developed a space radiation simulator, named the “GCR Simulator”, which allows for the more accurate radiobiologic research into the development and mitigation of radiation-induced malignancies [139]. Preflight colorectal cancer screening via colonoscopy or inflight screening via gut microbiome monitoring may be beneficial, but further research is required to demonstrate their clinical utility. Several studies, including the NASA Twins study have shown that microgravity could lead to alterations in an individual’s gut microbial community (i.e., gut dysbiosis) [2,140,141,142]. While changes to an individual’s gut microbiome can cause inflammation of the gastrointestinal tract [143,144], it remains unclear whether the specific alterations observed during spaceflight pose a risk to astronaut health. In fact, increased gut colonization by certain bacterial species is even associated with a beneficial effect on the gastrointestinal tract [2,140]. (Appendix A) Certain limitations of these studies, such as variations in genomic profile, diet, and a lack of adjusted confounders (e.g., the microbial content of samples) should be considered. Another potential consequence of prolonged microgravity exposure is the possibility of increased fatty-acid processing [145], leading to the development of non-alcoholic fatty liver disease (NAFLD) and hepatic fibrosis [146,147].

Further research is required to better understand gut microbial dynamics during space travel, as well as spaceflight-associated risk factors for the development of NAFLD, cholecystitis, and appendicitis.

### 1.4. Effects on the Immune System

During spaceflight, exposure to microgravity could potentially induce modifications in the cellular function of the human immune system. For example, it has been hypothesized that microgravity exposure could lead to an increase in the production of inflammatory cytokines [148] and stress hormones [149,150], alterations in the function of certain cell lines (NK cells [151,152], B cells [153], monocytes [154], neutrophils [154], T cells [5,155]), and impairments of leukocyte distribution [156] and proliferation [155,157,158]. The resultant immune system dysfunction could lead to the reactivation of latent viruses such as Epstein-Bar Virus (EBV), Varicella-Zoster Virus (VZV), and Cytomegalovirus (CMV) [31,32]. Persistent low-grade pro-inflammatory responses microgravity could lead to space fever. [159] Studies are currently underway to evaluate countermeasures to improve immune function and reduce reactivation of latent herpesviruses [33,160,161,162]. Microgravity exposure could also lead to the development of autoantibodies, predisposing astronauts to various autoimmune conditions [136,163]. (Appendix A) Most importantly, studies have shown that bacteria encountered within the space environment appear to be more resistant to antibiotics and more harmful in general compared to bacteria encountered on Earth [164,165]. This is in addition to the threat of novel bacteria species (e.g., *Methylobacterium ajmalii* sp. Nov. [76]) that we have not yet discovered.

Upon returning from the space environment astronauts remain in an immunocompromised state, which has been particularly problematic in the era of the COVID-19 pandemic. Recently, NASA has recommended postflight quarantine and immune status monitoring (i.e., immune-boosting protocol) to mitigate the risk of infection [77]. This is similar to the Apollo and NASA Health Stabilization Programs that helped establish the preflight protocol (pre-mission quarantine) currently used for this purpose.

Further research is required to understand the mechanisms of antibiotic resistance and the modifications in inflammatory cytokine dynamics, in order to develop immune boosters and surrogate immune biomarkers.

### 1.5. Effects on the Hematologic System

During short-duration spaceflight, the plasma volume and total blood volume de-crease within the first hours and remain reduced throughout the inflight period, a finding previously identified as space anemia [166]. Space anemia during spaceflight is perhaps due to a normal physiologic adaptation of newly released blood cells and iron metabolism to microgravity [167].

During long-duration spaceflight, microgravity exposure could potentially induce hemoglobin degradation, leading to hemolytic anemia. In a recent study of 14 astronauts who were on 6-month missions onboard the ISS, a 54% increase in hemolysis was ob-served after landing one year later [50]. In another small study, nearly half of astronauts (48%) landing after long duration missions were anemic and hemoglobin levels were characterized as having a dose–response relationship with microgravity exposure [51]. An additional study collected whole blood sample from astronauts during and after up to 6 months of orbital spaceflight [168]. Upon analysis, once the astronauts returned to Earth RBC and hemoglobin levels were significantly elevated. It is worth noting that these studies analyzed blood samples from astronauts collected after spaceflight, which may be influenced by various factors (e.g., the stress of landing and re-adaptation to conditions on Earth). In addition, these studies may be confounded by other extraterrestrial environmental factors such as fluid shifts, dehydration, and alteration of the circadian cycle.

Further research is urgently needed to understand plasma volume physiology dur-ing spaceflight and delineate the etiology and degree of hemolysis with longer space exposure, such as 1-year ISS or Mars exploration missions.

### 1.6. Oncologic Effects

Even during short-duration spaceflight, the stochastic nature of cancer development makes it possible that space radiation exposure could cause cancer via epigenomic modifications [63]. Currently, our epidemiological understanding of radiation-induced cancer risk is based primarily on atomic bomb survivors and accidental radiation exposures, which both show a clear association between radiation exposure and cancer risk [169,170]. However, these studies are hard to generalize to spaceflight as the patient populations vary significantly (generally healthy astronauts vs. atomic bomb survivors [NCRP 126]) [171]. Moreover, the radiation encountered in space is notably different than that associated with atomic bomb exposure. Most terrestrial exposures are based on low LET radiation (e.g., atomic bomb survivors received <1% dose from high LET neutrons) [172], whereas space radiation is comprised of higher LET ions (solar energetic particles and galactic cosmic rays) [173,174].

During long-duration spaceflight, our current understanding of cancer risk is also largely unknown. Our current epidemiologic understanding of long-duration radiation exposure and cancer risk is primarily based on the study of chronic occupational exposures and medically exposed individuals, supplemented with data obtained from animal studies, which are again based overwhelmingly on low LET radiation [169,170,175,176]. In animal studies, exposure to ionizing radiation (up to 13.5 months) has been associated with an increased risk of developing a variety of cancers [162,177,178,179,180]. Ionizing radiation exposure may cause DNA methylation patterns similar to the specific patterns observed in human adenocarcinomas and squamous cell carcinomas [63]; however, this response is not yet certain [181,182].

For the purposes of risk prediction, the elevated biological potency of heavy ions is modeled through concepts such as the radiation weighting factor, with NASA recently releasing unique quality factors (*Q_NASA_*) focused on high density tracks [183]. Although these predictive models can only estimate the impact of radiation exposure, extrapolation of current terrestrial-based data suggest that this risk could be at least substantial for astronauts. NASA, for example, has updated crew permissible career exposure limits to 0.6Sv, independent of age and sex. This degree of exposure results in a 2–3% mean increased risk of death from radiation carcinogenesis (NCRP 2021) [184]. This limit would be reached between 200 and 400 days of space travel (depending on degree of radiation shielding) [48].

Further research is urgently needed to understand the true risk of space radiation exposure. This is especially important for individuals with certain genotype-phenotype profiles (e.g., *BRCA1* or DNA methylation signatures) who may be more sensitive to the effects of radiation exposure. Most importantly, the utilization of genotype-phenotype profiles of astronauts or space travelers is valuable not just for pre-flight screening, but also during in-flight travel, especially for long-duration flights to deeper space. An individual’s genetic makeup will in-variably change during spaceflight due to the shifting epigenetic microenvironment. Future crewed-missions to deep space will have to adapt to these anticipated changes, be-come aware of impending red-flag situations, and determine whether any meaningful shift or change to ones’ genetic makeup is possible. For example, personalized radiation shields could potentially be tailored to an individuals’ genotype-phenotype profile, individualized pulmonary capillary wedge pressure under microgravity may be different due to transient changes in left atrial structure, or preflight analysis of the globin gene for the prediction of space anemia [50,129,185]. This research should be designed to identify the radiation type, dose, quality, frequency, and duration of exposure required for cancer development.

### 1.7. Effects on the Neurologic System

During the initial days of spaceflight, space motion sickness (SMS) is the most commonly encountered neurologic condition. Microgravity exposure during spaceflight commonly leads to alterations in spatial orientation and gaze stabilization (e.g., shape recognition [186], depth perception and distance [187,188]). Postflight, impairments in object localization during pitch and roll head movements [189,190] and fine motor control (e.g., force modulation [191], keyed pegboard completion time [192], and bimanual coordination [193]) are common. Anecdotally, astronauts also reported alterations in smell and taste sensations during their missions [27,194,195]. The observed impairment in olfactory function is perhaps due to elevated intracranial pressure (ICP) with increased cerebrospinal fluid outflow along the cribriform plate pathways [196]. However, to date, there have been no studies directly measuring ICP during spaceflight.

Upon returning from spaceflight, studies have observed that astronauts experience decrements in postural and locomotor control that can increase fall risk [197]. These decrements have been observed in both standard sensorimotor testing and functional tasks. While recovery of sensorimotor function occurs rapidly following short-duration spaceflight (within the first several days after return) [192,198], recovery after long-duration spaceflight often takes several weeks. Similar to SMS, post-flight motion sickness (PFMS) is very common and occurs soon after g-transition [30]. Deficits in dexterity, dual-tasking, and vehicle operation [199] are also commonly observed immediately after spaceflight. Therefore, short-duration astronauts are recommended to not drive automobiles for several days, and only after a sensorimotor evaluation (similar to a field sobriety test).

Similarly to the effects seen following short-duration spaceflight, those returning from long-duration spaceflight can also experience deficits in dexterity, dual-tasking, and vehicle operation. Long-duration astronauts are recommended to not drive automobiles for several weeks, and also require a sensorimotor evaluation. While central nervous system (CNS) changes [53] associated with long-duration spaceflight are commonly observed, the resulting effects of these changes both during and immediately after spaceflight remain unclear [199]. Observed CNS changes include structural and functional alterations (e.g., upward shift of the brain within the skull [54], disrupted white matter structural connectivity [55], increased fluid volumes [56], and increased cerebral vasoconstriction [200]), as well as modifications to adaptive plasticity [53]. Adaptive reorganization is primarily observed in the sensory systems. For example, changes in functional connectivity during plantar stimulation have been observed within sensorimotor, visual, somatosensory, and vestibular networks after spaceflight [201]. In addition, functional responses to vestibular stimulation were altered after spaceflight―reducing the typical deactivation of somatosensory and visual cortices [202]. These studies provide evidence for sensory reweighting among visual, vestibular, and somatosensory inputs.

Further research is required to fully understand the observed CNS changes. In addition, integrated countermeasures are needed for the acute effects of g-transitions on sensorimotor and vestibular function.

#### 1.7.1. Effects on the Neuro-Ocular System

Prolonged exposure to ionizing radiation is well known to produce secondary cataracts [61,62]. Most importantly, Spaceflight Associated Neuro-Ocular Syndrome (SANS) is a unique constellation of clinical and imaging findings which occur to astronauts both during and after spaceflight, and is characterized by: hyperopic refractive changes (axial hyperopia), optic disc edema, posterior globe flattening, choroidal folds, and cotton wool spots [43]. Ophthalmologic screening for SANS, including both clinical and imaging assessments is recommended. (Figure 2) Although the precise etiology and mechanism for SANS remain ill-defined, some proposed risk factors for the development of SANS include microgravity related cephalad fluid shifts [203], rigorous resistive exercise [204], increased body weight [205], and disturbances to one carbon metabolic pathways [206]. Many scientists believe that the cephalad fluid shift secondary to microgravity exposure is the major pathophysiological driver of SANS [203]. Although inflight lumbar puncture has not been attempted, several mildly elevated ICPs have been recorded in astronauts with SANS manifestations upon returning to Earth [43]. Moreover, changes to the pressure gradient between the intraocular pressure (IOP) and ICP (the translaminar gradient) have been proposed as a pathogenic mechanism for SANS [207]. The translaminar gradient may explain the structural changes seen in the posterior globe such as globe flattening and choroidal folds [207]. Alternatively, the microgravity induced cephalad fluid shift may impair venous or cerebrospinal drainage from the cranial cavity and/or the eye/orbit (e.g., choroid or optic nerve sheath). Impairment of the glymphatic system has also been proposed as a contributing mechanism to SANS, but this remains unproven [208,209]. Although permanent visual loss has not been observed in astronauts with SANS, some structural changes (e.g., posterior globe flattening) may persist and have been documented to remain for up to 7 years of long-term follow-up [210]. Further research is required to better understand the mechanism of SANS, and to develop effective countermeasures prior to longer duration space missions.

#### 1.7.2. Effects on the Neuro-Behavioral System

The combination of mission-associated stressors with the underlying confinement and social isolation of space travel has the potential to lead to cognitive deficits and the development of psychiatric disorders [211]. Examples of previously identified cognitive deficits associated with spaceflight include impaired concentration, short-term memory loss, and an inability to multi-task. These findings are most evident during G-transitions, and are likely due to interactions between vestibular and cognitive function [212,213]. Sopite syndrome, a neurologic component of motion sickness, may account for some cognitive slowing. The term “space fog”, has been used to describe the generalized lack of focus, altered perception of time, and cognitive impairments associated with spaceflight, which can occur throughout the mission. This may be related to chronic sleep deprivation as deficiencies (including decreased sleep duration and quality of sleep) are prevalent despite the frequent use of sleep medications [71]. These results highlight the broad impact of space travel on cognitive and behavioral health, and support the need for integrated countermeasures for long-duration explorative missions.

### 1.8. Effects on the Musculoskeletal System

During short-duration spaceflight, low back pain and disk herniation are common due to the presence of microgravity. While the pathogenesis of space-related low back pain and disk herniation is complex, the etiology is likely multifactorial in nature (e.g., microgravity induced hydration and swelling of the vertebral disk, muscle atrophy of the neck and lower back) [19,214,215]. Additionally, various joint injuries (e.g., space-suited shoulder injuries) can also occur in space due to the presence of microgravity [16,216,217,218]. Interestingly, one study showed that performing specific exercises could potentially promote automatic and tonic activation of lumbar multifidus and transversus abdominis as well as prevent normal lumbopelvic positioning against gravity following bed rest as a simulation of space flight [219], and the European Space Agency suggested that exercise program could relieve low back pain during spaceflight [220]. Further longitudinal studies are required to develop specialized exercise protocols during space travel.

During long-duration spaceflight, the presence of microgravity could cause an alteration in collagen fiber orientation within tendons, reduce articular cartilage and meniscal glycosaminoglycan content, and impair the wound healing process [22,23,24,221]. These findings seen in animal studies suggest that mechanical loading is required in order for these processes to occur in a physiologic manner. It is theorized that there is a mandatory threshold of skeletal loading necessary to direct balanced bone formation and resorption during healthy bone remodeling [222,223]. Despite the current countermeasure programs, the issue of skeletal integrity is still not solved [224,225,226].

Space radiation could also impact bone remodeling, though the net effect differs based on the amount of radiation involved [6]. In summary, high doses of space radiation lead to bone destruction with increased bone resorption and reduced bone formation, while low doses of space radiation actually have a positive impact with increased mineralization and reduced bone resorption. Most importantly, space radiation, particularly solar particle events in the case of a flare, may induce acute radiation effects, leading to hematopoietic syndrome [7]. This risk is highest for longer duration missions, but can be substantially minimized with current spacecraft shielding options.

Longitudinal studies are required to develop special exercise protocols and further assess the aforementioned risk of space radiation on the development of musculoskeletal malignancies.

### 1.9. Effects on the Pulmonary System

During short-duration spaceflight, a host of changes to normal, physiologic pulmonary function have been observed [73,227]. Studies during parabolic flight have shown that the diaphragm and abdomen are displaced cranially due to microgravity, which is accompanied by an increase in the diameter of the lower rib cage with outward movement. Due to the observed changes to the shape of the chest wall, diaphragm, and abdomen, alterations to the pressure-volume curve resulted in a net reduction in lung volumes [228]. In five subjects who underwent 25 s of microgravity exposure during parabolic flight, functional residual capacity (FRC) and vital capacity (VC) were found to be reduced [229]. During the Spacelab Life Sciences-1 mission, microgravity exposure resulted in 10%, 15%, 10–20%, and 18% reductions in VC, FRC, expiratory reserve volume (ERV), and residual volume (RV), respectively, compared to values seen in Earth’s gravity [227]. The observed physiologic change in FRC is primarily due to the cranial shift of the diaphragm and abdominal contents described previously, and secondarily to an increase in intra-thoracic blood volume and more uniform alveolar expansion [227].

One surrogate measure for the inhomogeneity of pulmonary perfusion can be assessed through changes in cardiogenic oscillations of CO_2_ (oscillations in exhaled gas composition due to differential flows from different lung regions with differing gas composition). Following exposure to microgravity, the size of cardiogenic oscillations were significantly reduced to 60% in comparison to the preflight standing values [230,231]. Possible causes of the observed inhomogeneity of ventilation include regional differences in lung compliance, airway resistance, and variations in motion of the chest wall and diaphragm. Access to arterial blood gas analysis would allow for enhanced physiologic evaluations, as well as improved management of clinical emergencies (e.g., pulmonary embolism) occurring during space travel. However, there is currently no suitable method for assessing arterial blood in space. The earlobe arterialized blood technique for collecting blood gas has been proposed, but evidence is limited [232]. Further research is required in this area to establish an effective means for sampling arterial blood during spaceflight.

In comparison to the changes seen during short-duration spaceflight, studies conducted during long-duration spaceflight showed that the heterogeneity of ventilation/perfusion (V/Q) was largely unchanged, with preserved gas exchange, VC, and respiratory muscle strength [73,233,234]. This resulted in overall normal lung function. This is supported by long-duration studies (up to 6 months) in microgravity which demonstrated that the function of the normal human lungs is largely unchanged following the removal of gravity [233,234]. It is worth noting that there were some small changes which were observed (e.g., an increase in ERV in the standing posture) following long-duration spaceflight, which can perhaps be attributed to a reduction in circulating blood volume [233,234]. However, while microgravity can causes temporary changes in lung function, these changes were reversible upon return to Earth’s gravity (even after 6 months of exposure to microgravity). Based on the currently available data, the overall effect of acute and sustained exposure to microgravity does not appear to cause any deleterious effects to gas exchange in the lungs. However, the biggest challenge for long-duration spaceflight is perhaps extraterrestrial dust exposure. Further research is required to identify the long term consequences of extraterrestrial dust exposure and develop potential countermeasures (e.g., specialized face masks) [73].

### 1.10. Effects on the Dermatologic System

During short-duration space travel, skin conditions such as contact dermatitis, skin sensitivity, biosensor electrolyte paste reactions, and thinning skin are common [44,235]. However, these conditions are generally mild and unlikely to significantly impact astronaut safety or prevent completion of space missions [44].

The greatest dermatologic concern for long-duration space travelers is the theoretical increased risk of developing skin cancer due to space radiation exposure. This hypothesis is supported by one study which found the rate of basal cell carcinoma, melanoma, and squamous cell carcinoma of the skin to be higher among astronauts compared to a matched cohort [236]. While the three-fold increase in prevalence was significant, there were a number of confounders (e.g., the duration of prolonged UV exposure on Earth for training or recreation, prior use of sunscreen protection, genetic predisposition, and variations in immune system function) that must also be taken into account. A potential management strategy for dealing with various skin cancers during space travel involves telediagnostic and telesurgical procedures. Further research is needed to improve the telediagnosis and management of dermatological conditions (e.g., adjustment for a lag in communication time) during spaceflight.

### 1.11. Diagnostic Imaging Modalities in Space

In addition to routine physical examination, various medical imaging modalities may be required to monitor and diagnose medical conditions during long-duration space travel. To date, ultrasound imaging acquired on space stations has proven to be helpful in diagnosing a wide array of medical conditions, including venous thrombosis, renal and biliary stones, and decompression sickness [29,237,238,239,240,241,242]. Moreover, the Focused Assessment with Sonography for Trauma (FAST), utilized by physicians to rapidly evaluate trauma patients, may be employed during space missions to rule out life-threatening intra-abdominal, intra-thoracic, or intra-ocular pathology [243]. Remote telementored ultrasound (aka tele-ultrasound) has been previously investigated during the NASA Extreme Environment Missions Operations (NEEMO) expeditions [244]. Today, the Butterfly iQ portable ultrasound probe can be linked directly to a smartphone through cloud computing, allowing physicians/specialists to promptly analyze remote ultrasound images [245].

Currently, alternative imaging modalities such as X-ray, CT, PET and MRI scan are unable to be used in space due to substantial limitations (e.g., limited space for large imaging structures, difficulties in interpretation due to microgravity). However, it is possible that the future development of a photocathode-based X-ray source may one day make this a possibility [101,246]. If X-ray imaging was possible, certain caveats would need to be taken into account for accurate interpretation. For example, pleural effusions, air-fluid levels, and pulmonary cephalization commonly seen on terrestrial imaging, would need to be interpreted in an entirely different way due to the effect of microgravity [247]. While this adjustment might be challenging, the altered principles of weightless physiology may provide some advantages as well. For example, one study found that intra-abdominal fluid was better able to be detected in space than in the terrestrial environment due to gravitational alterations in fluid dynamics [248]. Further research is required to identify and optimize inflight imaging modalities for the detection and treatment of various medical conditions.

### 1.12. Medical and Surgical Procedures in Space

Despite the presence of microgravity, both basic life support and advanced cardiac life support are feasible during space travel with some modifications [249,250]. For example, the recent guidelines for CPR in microgravity recommend specialized techniques for delivering chest compressions [251]. The use of mechanical ventilators, and moderate sedation or general anesthesia in microgravity are also possible but the evidence is extremely limited [252,253]. In addition, there are several procedures such as endotracheal intubation, percutaneous tracheostomy, diagnostic peritoneal lavage, chest tube insertion, and advanced vascular access which have only been studied through artificial stimulation [254,255].

Once traditionally “surgical” conditions are appropriately diagnosed, the next step is to determine whether these conditions should be managed medically, percutaneously, or surgically (laparoscopic vs. open procedures) [47,256]. For example, acute appendicitis or cholecystitis that would historically be managed surgically in terrestrial hospitals, could instead be managed with antibiotics rather than surgery. While the use of antibiotics for these conditions is usually effective on Earth, there remain concerns due to space-induced immune alterations, increased pathogenicity and virulence of microorganisms, and limited resources to “rescue” cases of antibiotic failure [39]. In cases of antibiotic failure, one potential minimally invasive option could be ultrasound-guided percutaneous drainage, which has previously been demonstrated to be possible and effective in microgravity [257]. Another potential approach is to focus on the early diagnosis and minimally invasive treatment of appropriate conditions, rather than treating late stage disease. In addition to expediting the patient’s post-operative recovery, minimally invasive surgery in space has the added benefit of protecting the cabin environment and the remainder of the crew [258,259].

As in all aspects of healthcare delivery in space, the presence of microgravity can complicate even the most basic of procedures. However, based on collective experience to date, if the patient, operators, and all required equipment are restrained, the flow of surgical procedures remains relatively unchanged compared to the traditional, terrestrial experience [260]. A recent animal study confirmed that it was possible to perform minor surgical procedures (e.g., vessel and wound closures) in microgravity [261]. Similar study during parabolic flight has further confirmed that emergent surgery for the purpose of “damage control” in catastrophic scenarios can be conducted in microgravity [262]. As discussed previously, telesurgery may be feasible if the surgery can be performed with an acceptably brief time lag (<200 ms) and if the patient is within a low Earth orbit [263,264]. However, further research and technological advancements are required for this to come to fruition.

### 1.13. Lifestyle Management in Space

Based on microgravity simulation studies, NASA has proposed several potential biomedical countermeasures in space [33,160,161]. Mandatory exercise protocols in space are crucial and can be used to maintain physical fitness and counteract the effects of microgravity. While these protocols may be beneficial, exercise alone may not be enough to prevent certain effects of microgravity (e.g., an increase in arterial thickness/stiffness) [20,265,266,267]. For example, a recent study found that resistive exercise alone could not suppress the increase in bone resorption that occurs in space [20]. Hence, a combination of resistance training and an antiresorptive medication (e.g., bisphosphonate) appears to be optimal for promotion of bone health [20,21]. Further research is needed to identify the optimal exercise regimen including recommended exercises, duration, and frequency.

In addition to exercise, dietary modification may be another potential area for optimization. The use of a diet based on caloric restriction (CR) in space remains up for debate. Based on data from terrestrial studies, caloric restriction may be useful for improving vascular health; however, this benefit may be offset by the associated muscle atrophy and osteoporosis [268,269]. Given that NASA encourages astronauts to consume adequate energy to maintain body mass, there has been an attempt to mimic the positive effects of CR on vascular health while providing appropriate nutrition. Further research is needed this area to identify the ideal space diet.

Based on current guidelines, only vitamin D supplementation during space travel is recommended. Supplementation of A, B6, B12, C, E, K, Biotin, folic acid are not generally recommended at this time due to insufficient evidence [64] (Appendix A). The use of traditional prescription medications may not function as intended on Earth. Therefore, alternative methods such as synthetic biologic agents or probiotics may be considered [35,38]. However, evidence in this area is extremely limited, and it is possible that the synthetic agents or probiotics may themselves be altered due to microgravity and radiation exposure. Further research is needed to investigate the relationship between these supplements and potential health benefits in space.

Currently, most countermeasures are directed towards cardiovascular system and musculoskeletal pathologies but there is little data against issues like immune and sleep deprivation, SANS, skin, etc. Artificial Gravity (AG) has been postulated as adequate multi-system countermeasure especially the chronic exposure in a large radius systems. Previously, the main barrier is the huge increase in costs [270,271,272]. However, there are various studies that show the opposite and also the recent decrease in launch cost makes the budget issue nearly irrelevant especially when a huge effort is paid to counteract the lack of gravity. The use of AG especially long-radius chronic AG is feasible. Further studies are needed to determine the utilization of AG in long-duration space travel.

### 1.14. Future Directions for Precision Space Health with AI

In this new era of space travel and exploration, ‘future’ tools and novel applications are needed in order to prepare deep space missions, particularly pertaining to strategies for mitigating extraterrestrial environmental factors, including both exogenous and en-dogenous processes. Such ‘future’ tools could help assist and ensure a safe travel to deep space, and more importantly, help bring space travelers and astronauts back to Earth. These tools and methods may initially be ‘remotely’ controlled, or have its data sent back to Earth for analysis. Primarily efforts should be focused on analyzing data in situ, and on site during the mission itself, both for the purpose of efficiency, and for the progressive purpose of slowly weaning off a dependency on Earth.

AI is an emerging tool in the big data era and AI is considered a critical aspect of ‘fu-ture’ tools within the healthcare and life science fields. A combination of AI and big data can be used for the purposes of decision making, data analysis and outcome prediction. Just recently, there have been encourage in advancements in AI and space technologies. To date, AI has been employed by astronauts for the purpose of space exploration; however, we may just be scratching the surface of AI’s potential. In the area of medical research, AI technology can be leveraged for the enhancement of telehealth delivery, improvement of predictive accuracy and mitigation of health risks, and performance of diagnostic and interventional tasks [273]. The AI model can then be trained and have its inference leveraged through cloud computing or Edge TPU or NVIDIA Jetson Nano located on space stations. (Table 3 and Appendix A) Figure 3 demonstrates potential AI applications in space.

As described previously, the capability to provide telemedicine beyond LEO is primarily limited by the inability to effectively communicate between space and Earth in real-time [274]. However, AI integration may be able to bridge the gap and advance communication capabilities within the space environment [275,276]. One study demonstrated a potential mechanism for AI incorporation in which an AI-generated predictive algorithm displayed the projected motion of surgical tools to adjust for excess communication lag-time [277]. This discovery could potentially enable AI-enhanced robotics to complete repetitive, procedural tasks in space without human inputs (e.g., vascular access) [278]. Today, procedures performed with robotic assistance are not yet fully autonomous (they still require at least one human expert). It is possible with future iterations that an AI integration could be created with the ability to fully replicate the necessary human steps to make terrestrial procedures (e.g., percutaneous coronary intervention, incision and drainage [103], telecholecystectomy [105,106], etc.) feasible in space [275,279]. The seventh NEEMO mission previously demonstrated that robotic surgery controlled by a remote physician is feasible within the environment of a submarine, but it remains to be seen whether this can be expanded to the space environment [280].

On space stations, Edge TPU-accelerated AI inference could be used to generate accurate risk prediction models based on data obtained from simulated environments (e.g., NASA AI Risk Prediction Challenge) [281]. For example, AI could potentially utilize data (e.g., -omics) obtained from research conducted both on Earth and in simulated environments (e.g., NASA GCR Simulator) to predict an astronaut’s risk of developing cancer due to high-LET radiation exposure (cytogenetic damage, mitochondrial dysregulation, epigenetic alterations, etc.) [63,78,79,282,283,284].

Another potential area for AI application is through integration with wearable technology to assist in the monitoring and treatment of a variety of medical conditions. For example, within the field of cardiovascular medicine, wearable sensor technology has the capability to detect numerous biosignals including an individual’s cardiac output, blood pressure, and heart rate [285]. AI-based interpretation of this data can facilitate prompt diagnosis and treatment of congestive heart failure and arrhythmias [285]. In addition, several wearable devices in various stages of development are being created for the detection and treatment of a wide array of medical conditions (obstructive sleep apnea, deep vein thrombosis, SMS, etc.) [285,286,287,288].

As discussed previously, the confinement and social isolation associated with prolonged space travel can have a profound impact on an astronaut’s mental health [8,10,67]. AI-enhanced facial and voice recognition technology can be implemented to detect the early signs of depression or anxiety better than standardized screening questionnaires (e.g., PHQ-9, GAD-7) [68,69]. Therefore, telepsychology or telepsychiatry can be used pre-emptively for the diagnosis of mental illness [68,69,289].

## 2. Conclusions

Over the next decade, NASA, Russia, Europe, Canada, Japan, China, and a host of commercial space companies will continue to push the boundaries of space travel. Space exploration carries with it a great deal of risk from both known (e.g., ionizing radiation, microgravity) and unknown risk factors. Thus, there is an urgent need for expanded research to determine the true extent of the current limitations of long-term space travel and to develop potential applications and countermeasures for deep space exploration and colonization. Researchers must leverage emerging technology, such as AI, to advance our diagnostic capability and provide high-quality medical care within the space environment.

## Figures and Tables

**Figure 1 cells-12-00040-f001:**
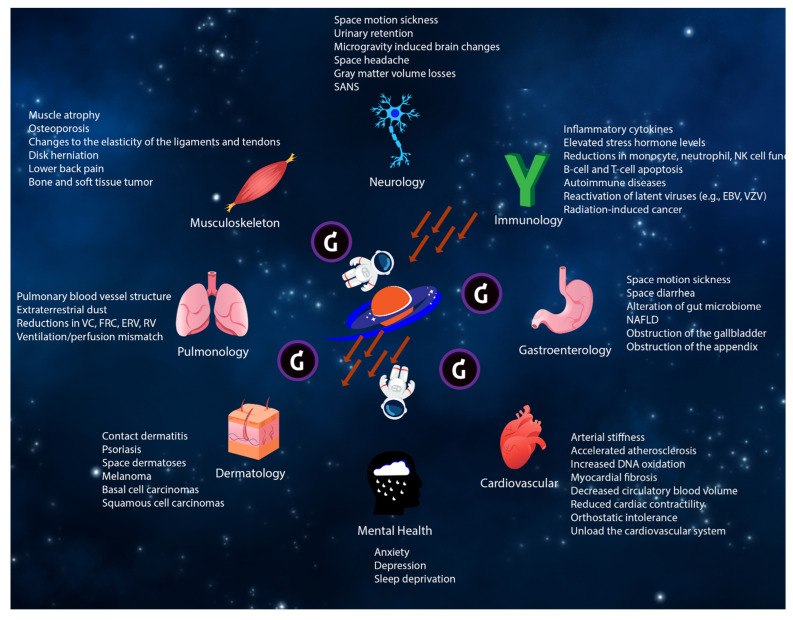
Potential effects of the space environment on each organ system.

**Figure 2 cells-12-00040-f002:**
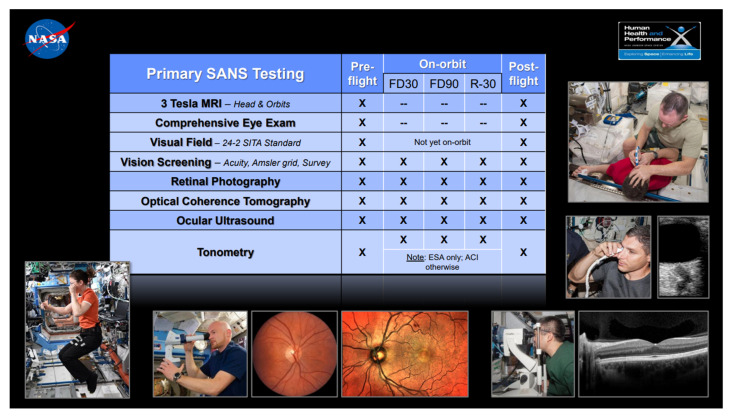
Ophthalmologic screening for SANS.

**Figure 3 cells-12-00040-f003:**
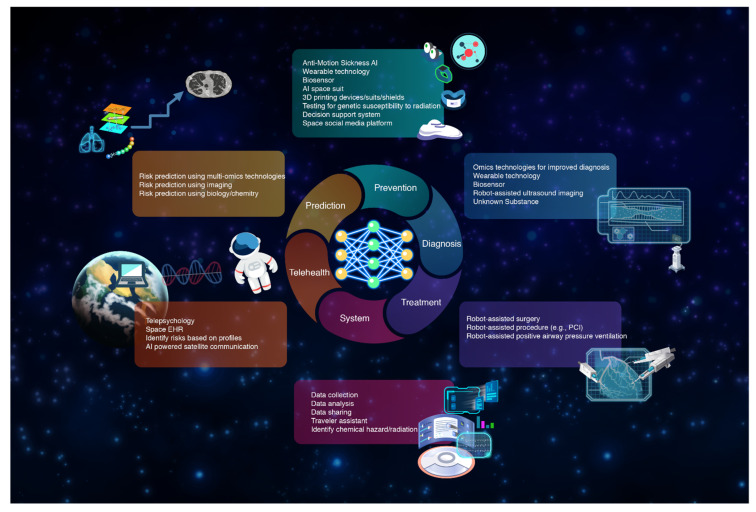
Potential AI applications in space.

**Table 1 cells-12-00040-t001:** Summarizes the pre-flight, in-flight and post-flight screening in each system.

Scheme	Pre-Flight	In-Flight	Post-Flight
Physical Fitness Assessment	Cardiovascular fitness: VO_2max_ measured cycle ergometry Flexibility: Sit & reach; shoulder flexibility Muscular strength and endurance: Maximum push-ups in 2 min; maximum sit-ups in 2 min; maximum pull-ups; handgrip strength with dynamometer	Cardiovascular fitness: Flight day (FD) +30, +120, return day (R) −30 VO_2max_ FD+3 through day prior to return. Resistance: 3x/wk (60 min) Aerobic: 3x/wk (30 min)	Functional fitness assessment: R+7, R+30 Cardiovascular fitness: R+7, R+30 VO_2max_
Cardiovascular	Coronary artery calcium (CAC) score every 5 years Echocardiogram annually VO_2max_ cycle test annually Fasting lipid panel High Sensitivity C-Reactive Peptide (HS-CRP) Carotid Intima-Media Thickness (CIMT) annually Individuals with cardiovascular disease history (e.g., compensated heart failure, asymptomatic CAD, arrhythmias) are not currently eligible for either long- or short-term space travel	Quarterly VO_2max_ cycle testing Quarterly vascular US Monthly health surveillance exam with blood pressure measurement	VO_2max_ cycle test at R+7, R+30 Resting 12-lead ECG annually Blood pressure measurement Echocardiogram Fasting lipid panel HS-CRP Hypertension screening using a sphygmomanometer annually and as clinically indicated Cardiovascular health screening annually
Pulmonology	Pulmonary function testing annually 6-Minute Walk Test for patients with chronic pulmonary disease	None	Pulmonary function testing Purified Protein Derivative (PPD) (tuberculin) skin test for tuberculosis screening, unless clinically contraindicated
Ophthalmology	SANS-surveillance, to include baseline testing of: Visual acuity OCT US MRI orbits Tonometry Ophthalmologic examination Color vision Phoria testing Ophthalmologic screening for ocular health and visual status (e.g., testing for refractive errors or cataracts) is recommended	See attached chart for inflight scheduling details	Visual acuity, color vision, and extraocular muscle testing (annually) Tonometry (annually) Dilated fundus exam, retinal photographs, and optical coherence tomography (OCT)
Otorhinolaryngology	Screening audiometry Tympanography	Quarterly On Orbit Hearing Assessment (OOHA)	Hearing questionnaire and pure-tone audiometry testing (annually)
Hematology	Complete Blood Count (CBC) with differential Iron studies	CBC every 60 days	CBC, CMP (annually)
Immunology	Administration of vaccines (e.g., Shingrix) for prevention of inflight VZV reactivation Immunoglobulins (annually) TB screening with PPD, Quantiferon Gold	None	NASA has recommended post-flight quarantine and immune status monitoring (i.e., immune-boosting protocol) to mitigate the risk of infection
Gastroenterology	Age appropriate colon cancer screening	None	Lifetime Surveillance of Astronaut Health (LSAH)
Musculoskeletal	Annual DXA for osteoporosis screening for postmenopausal females and males >50 years old Cervical and thoracic spine MRI	None	Post-flight DXA for BMD Spine MRI
Dermatology	Annual skin examination	None	Visual exam of the skin with photo documentation of any abnormalities, particularly for melanoma and non-melanoma skin cancers (R+0/1 day and annually)
Genitourinary	PSA in males Annual urinalysis, 24-h urine collection	Quarterly 24-h urine collection	Urinalysis 24-h urine collection
Neurology	MRI brain Neurovestibular platform test	None	Neurovestibular platform test
Endocrine	TSH, free T4 Thyroid US	None	Annual TSH, free T4
Dental	Annual dental examinations Orthopantomogram X-rays	None	Post-flight dental examination
Oncology	Age-appropriate pap smear and mammogram in females Age-appropriate colon cancer screening		

**Table 2 cells-12-00040-t002:** Summary of Space Hazards to each organ system and potential countermeasures.

Tread/Hazards	Health Risks	Current Countermeasure
**Microgravity**		
Musculoskeletal	Muscle and bone atrophy [19]	A combination of adequate exercise and antiresorptive medications [20,21]
	Impaired wound healing [22,23,24]	No specific countermeasures exist
Neurology	Space motion sickness (SMS) [25]	Scopolamine administered orally or via transdermal patch with or without an amphetamine (to counteract drowsiness), or intramuscular Promethazine [26]
	Alterations of smell and taste [27]	No specific countermeasures exist
	Self-reported congestion [27]	Lower body negative pressure may help, but the effects are transient
	Internal jugular vein thrombosis [28,29]	No specific countermeasures exist
	Post-flight motion sickness [30]	No specific countermeasures exist
Immunology	Reactivation of latent viruses [31,32]	Preflight vaccination [32] Treatment with polyclonal immunoglobulin (IG) and Interleukin-2 (SC) [33] Adequate exercise and nutrition [33]
	Hypersensitivity [34]	No specific countermeasures exist
Gastroenterology	Gut microbiota dysbiosis [2]	Probiotics, synthetic peptides, synthetic biology [35,36,37,38]
	Acute appendicitis and cholecystitis	Inflight medical management or minimally invasive procedures (e.g., ultrasound guided percutaneous drainage) of traditionally “surgical” conditions. [39] Alternatively, can consider prophylactic minimally invasive surgery prior to prolonged space flight in at-risk individuals [40]
Cardiovascular	Orthostatic intolerance [41]	Regular exercise training and adequate hydration prior to landing
	Cardiac myocyte atrophy [42]	Restores back to baseline after returning from spaceflight
Ophthalmology	Spaceflight-Associated Neuro-Ocular Syndrome (SANS) [43]	The lower body negative pressure device has been proposed as a countermeasure but is currently unproven
Dermatology	Skin tinning, dry skin, delayed wound healing, or skin infections [44]	No specific countermeasures exist
Traumatology	Increased risk of traumatic injury with construction, inadvertent acceleration of large masses when in zero gravity, impaired physiologic responses to hemorrhage and shock in zero gravity, bone demineralization increases risk of traumatic fractures, micrometeorites in EVA activities [45,46,47]	No specific countermeasures exist
**Radiation**		
Most systems	Cancer induction	Shielding, including advanced shielding options. [48,49] Optimize mission according to solar cycle [48,49] Sex- and age-based selection criteria for astronauts [48]
Musculoskeletal	Bone cell remodeling [6]	Shielding [7]
Hematology	Acute hematopoietic effects [7]	Shielding [7]
	Space-related hemolytic anemia [50,51]	Screening and monitoring for hemolysis
Neurology	Neuro-degenerative disorders and central nervous system changes [52,53,54,55,56]	Adaptive visuo-motor training, active sensory feedback to guide task performance, galvanic vestibular stimulation (GVS)
Gastroenterology	Gut microbiota dysbiosis [2]	Probiotics, synthetic peptides, synthetic biology [35,36,37,38]
Cardiovascular	Accelerated atherosclerosis [3,4,57,58,59,60]	Shielding [7]
Ophthalmology	Radiation-induced cataract [61,62]	No specific countermeasures exist
Pulmonology	Lung adenocarcinomas and squamous cell carcinomas [63]	Shielding [7]
Dermatology	Melanoma and non-melanoma skin cancers [44]	Inflight telediagnostic system
**Others**		
Nutrition	Nutritional deficiencies [64]	Telehealth and routine lab work for identification of nutritional deficiencies, and adequate supplementation as needed
Mental health	Psychologic distress [10,65]	Stress-relieving breathing exercises and/or mindfulness/positive visualization exercises [66]
	Depression, confinement, isolation, immobilization lack of social interaction [10,65,67]	Telepsychiatry, telepsychotherapy, space social media using LunaNet [68,69,70].
Altered circadian rhythms	Circadian de-synchronization and sleep loss [71]	Melatonin supplementation may be useful [72]
Dust exposure	Extraterrestrial dust exposure [73]	No specific countermeasures exist
Auditory exposures	Noise generated from man-made sources such as mechanical equipment (Noise/sound does not propagate in the vacuum of space and it requires an atmosphere to propagate, e.g., the pressurized spacecraft interior) [74]	Hearing protection in the form of foam ear plugs [74]
Unknown infections	Mission-threatening risks from unknown infections or novel bacterial/viral phenotypes [75,76]	PPE and post-exposure quarantine [77]

**Table 3 cells-12-00040-t003:** Potential AI applications in space health.

Potential AI Applications
**Risk Prediction Models**
Risk prediction models for development of cancer due to ionizing radiation exposure based on animal models [78,79]
Inflight AI assisted multi-omic research using the Edge TPU to generate novel risk prediction scores [80]
Risk prediction models for the development of cardiovascular disease based on carotid-femoral pulse wave velocity obtained via Doppler ultrasound or AI-guided analysis of 3D printed tissue models [81]
Risk prediction models such as the modified Astro-CHARM model which combines a patient’s CAC score with non-traditional risk factors associated with space travel (e.g., radiation exposure, microgravity exposure, dysbiosis of gut microbiota, etc.) to identify those at risk for accelerated atherosclerosis [82]
Predicting antimicrobial resistance of novel bacterial species (e.g., *Methylobacterium ajmalii* sp. nov., *Methylobacterium rhodesianum,* etc.) based on whole genome sequencing data [83]
AI-based models for the identification of causal relationships between various exposures and subsequent cancer development (e.g., CRISP) [84]
**Medical Screening Preflight and Inflight Monitoring**
AI monitoring for gut microbiota dysbiosis using smart toilets during pre-, intra-, and post-flight periods [2,85,86]
Screening for latent virus infection or reactivation based on immunological variables, which can then be used as a biomarker for assessment of adaptive immune function [87,88]
The Edge TPU and other similar low-power AI ASICs could be used to monitor genomic imprinting (e.g., DNA methylation signatures or epigenetic/transcriptomic-based tests) for inflight lung cancer screening (may be equivalent to low-dose CT chest screening) [63,89,90]
AI screening for mental illness through facial recognition and vocal analysis [68,69]
A combination of AI and imaging could potentially identity abnormal or misfolded proteins [91,92]
Deep learning algorithms for the inflight telediagnosis of various conditions (e.g., skin lesions or SANS) [93,94,95]
AI-integrated wearable technology for space suits (e.g., exoskeletons) to monitor biodata (e.g., electrocardiogram, blood pressure, sleep cycle, body temperature, etc.) [96,97]
AI monitoring of metabolomics-based biomarkers for assessment of sleep quality and circadian rhythm [98]
AI can be used to analyze retinal fundus photography to identify and monitor space anemia [99]
**Medical Diagnostic Tools**
The Edge TPU and other similar low-power AI ASICs (i.e., Google Coral Edge TPU, NVIDIA Jetson Nano, Intel Neural Compute Stick 2) could advance image segmentation AI models, allowing for these models to be feasibly deployed in ultrasound point-of-care settings (e.g., detection of DVT, structural heart disease, hemodynamic changes), even procedural guidance [100].
AI-based system designed to guide non-physicians on the proper acquisition of medical diagnostic testing using The Edge TPU and deep reinforcement learning
AI-enhanced 3D-imaging technology (e.g., micro-CT scanners) [101]
Utilization of the Edge TPU and other similar low-power AI ASICs to provide the necessary processing power for high-performance parallel-processing space research [102]
**Intervention**
AI-guided minor surgical procedures [such as incision and drainage (I&D)] using next-generation High Performance Spaceflight Computing (HPSC) [103]
AI-assisted, remotely controlled robotic PCI and robotic laparoscopic surgery (e.g., telecholecystectomy and teleappendectomy); made possible by a reduction in communication latency beyond a lag of 200 ms [103,104,105,106,107]
Using AI predictions of drug metabolism and effectiveness based on an individual’s multiomic data prior to medication or supplement distribution (e.g., melatonin, immune supplements, probiotics) [108,109]
**Disease Prevention**
AI-integrated space suits (e.g., exoskeletons) to maximize EVA time and operating pressure, and minimize space radiation exposure [110,111]
3D printing of personalized devices (e.g., ear plugs to prevent noise source generated from man-made sources), space shields, space suits for use in emergency scenarios [112,113,114]
AI-based chatbots or social media could potentially be used to prevent anxiety and depression during long-duration space travel. The Edge TPU could potentially be used in advancing an internet or social media for the moon, known as LunaNet [115].
AI-improved augmented reality/virtual reality to reduce/prevent Spaceflight-Associated Neuro-Ocular Syndrome (SANS), space motion sickness (SMS), and post-flight motion sickness (PFMS) [116,117,118]
AI identification of an individual’s radiosensitivity (genotype-phenotype) using multiomics to prevent the negative effects of ionizing radiation [119,120,121]
Emerging technology could allow for the transfer of large volumes of data at faster rates in order to facilitate medical research in space [122]
AI can be used to create novel metrics (e.g., travelers’ satisfactions, post spaceflight measurements for travelers) and ethics (e.g., health insurance)

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
