# Peer review of "Human Health during Space Travel: State-of-the-Art Review"

_cells, 2022, doi:10.3390/cells12010040_

Round 1
Reviewer 1 Report
The Review "Human Health During Space Travel: State-of-the-Art"
by Chayakrit Krittanawong” et al. is very interesting.
The authors have provided a thorough and engaging review of the medical literature on human experiments conducted during space travel and summarize our current knowledge on the effects of living in space for both short and long durations of time.
The few aspects:
Unfortunately, an Abstract does not contain full information about the Manuscript. What knowledge have the authors gathered and what do they discuss?
Point 1
Please add an Abstract and the Keywords in the Manuscript.
Point 2. Figure 1 and 2
Could to increase the Text Size to increase the font on the pictures, as it is a little difficult to make out the captions.
In conclusion, this work will make an essential and significant contribution to space medicine and is a worthy example for writing scientific articles.
Author Response
Chayakrit Krittanawong, MD
Brian E Crucian, PhD
Center for Space Medicine - Baylor College of Medicine
NASA Johnson Space Center
December 4, 2022
Cells
Dear Editors,
We are pleased to resubmit the manuscript entitled “Human Health During Space Travel: State-of-the-Art Review” for publication in Cells for Special Issue "Research Advances Related to Cardiovascular System"
We have carefully reviewed the Editors’ and reviewers’ comments. Below, we provide a point by point response to these comments. The changes in the manuscript are noted as yellow color highlight.
The manuscript, as submitted or its essence in another version, is not under consideration for publication elsewhere and will not be published elsewhere while under consideration by the Cells. The authors have no commercial associations or sources of support that might pose a conflict of interest with this topic.
We are most appreciative of your consideration for publication in any type and look forward to your comments.
Sincerely yours,
Chayakrit Krittanawong, MD
Brian E Crucian, PhD
Editor’s comments
- Please do follow the pre-layout version of your manuscript (you can download it from our system) when you revise this manuscript, you will find the accurate line numbers, it's also beneficial for our editorial
work afterward.
Response: Thank you very much for suggestions. We revised the manuscript in the pre-layout version of your manuscript.
2)Please include the first ten authors' names before using “et al.” in the references. such as:Isobe, Y.; Okumura, M.; McGregor, L.M.; Brittain, S.M.; Jones, M.D.; Liang, X.; White, R.; Forrester, W.; McKenna, J.M.; Tal-larico, J.A.; et al. Manumycin polyketides act as molecular glues between UBR7 and P53. Nat. Chem. Biol. 2020, 16, 1189–1198 And the citation format of reference should be [1], [2,3],[4-7]......
Response: Thank you very much for suggestions. We reformatted references as suggested.
3)The main text needs to be numbered according to the following form:
1.
2.
2.1
2.2... ...
3.
3.1
... ...
Response: Thank you very much for suggestions. We reformatted the sections as suggested.
4) figure1 and 2 copyright are needed, please provide a citation in the following format: “Reprinted/adapted with permission from Ref. [XX]. Copyright year, copyright owner’s name”. Or reprinted with permission
from Author Names (Year of Publication). Copyright Year Copyright Owner’s Name. If you adopt or use only part of a figure or table, permission is still needed.
Response: Thank you very much for suggestions. The figure is exclusively made by myself and they are original as they have been made by myself.
Reviewers’ comments
The Review "Human Health During Space Travel: State-of-the-Art"
by Chayakrit Krittanawong” et al. is very interesting.
The authors have provided a thorough and engaging review of the medical literature on human experiments conducted during space travel and summarize our current knowledge on the effects of living in space for both short and long durations of time.
The few aspects:
Unfortunately, an Abstract does not contain full information about the Manuscript. What knowledge have the authors gathered and what do they discuss?
Response: Thank you very much for insightful suggestions. We have updated the abstract to reflect the entire manuscript contents.
Point 1
Please add an Abstract and the Keywords in the Manuscript.
Response: Thank you very much for suggestions. We have added abstract and keywords in the manuscript.
Point 2. Figure 1 and 2
Could to increase the Text Size to increase the font on the pictures, as it is a little difficult to make out the captions.
Response: Thank you very much for suggestions. We have revised the figures by magnifying the texts as suggested.
In conclusion, this work will make an essential and significant contribution to space medicine and is a worthy example for writing scientific articles.
Thank you very much for your thorough and thoughtful review. We have taken your recommendations under great consideration and hope that our revamped manuscript meets the high standard associated with Cells.
Dear Authors.
I would like to start by congratulating you for an extensive review and manuscript as the one you have submitted for publication consideration. I have read with great interest your review, and notice that it provides a very good and comprehensive review to the field. There are perhaps three points / comments that I'd like you to consider for revision. These include;
1) As an article / review focusing on health implications in space, and counter measures, you rightly mention hematology as one of the areas important. However, you did not mention the importance of space anemia, the destructions of RBCs in vivo in astronauts in space vs. those that remain on Earth. A quick search online, will turn up a number of important publications, and cited also in major media establishments (CNN, BBC...) At least one article was written by a group in Ottawa (Guy Trudel..) I think this merits mention and entry
Response: Thank you very much for insightful suggestions. We agree that hematology section is one of the areas important and added dedicated section “hematologic effects”
2) Another important point - is the use of (i) Genetic and (ii) DNA profiling - of astronauts not just before launch - but also the use of such tools during in-flight especially for long-duration flights to deeper space. The genetic makeup will invariably change to the shifting epigenetic microenvironment in-flight, future crewed-missions to deep space will have to adapt to these changes, become aware of impending red-flag situations, and whether any meaningful shift or change to ones' genetic makeup is at risk.
Response: Thank you very much for suggestions. We have added the important use of Genetic and DNA profiling as suggested.
and so for my third point,
3) Perhaps a very small section, not more than a paragraph entry, that details about 'future' tools and methods applications for deep space missions. How these will help, assist and ensure a safer trip to deeper space missions, and more importantly, how to bring these people back to earth. Such tools and methods may initially be 'remotely' controlled or have its data sent to earth for analysis, however effort and progress should be focused on analysing data in situ, and on site during the mission itself for efficiency, and progressive purposes and slowly weaning off dependency on Earth.
Response: Thank you very much for suggestions. We have added details about 'future' tools and methods applications for deep space missions as suggested.
Those three points, I feel - if added to your review - will strengthen the reader base, will make it more appealing, and attract a large number of Life Scientists interest' in the field of space medicine, and space bioscience.
Response: Thank you very much for your thorough and thoughtful review. We have taken your recommendations under great consideration and hope that our revamped manuscript meets the high standard associated with Cells.
The manuscript by Krittanawong and quite some other colleagues provides a wide overview of the effects of spaceflight on human health. List of co-authors is a plethora of space-flight related experts wide a wide experience in the field.
The title does not reflect the content of the manuscript with respect to the specific attention to the subject of AI.
Response: Thank you very much for suggestions.
L66: “Here, we review the available medical literature on human experiments conducted during space travel and summarize our current knowledge on the effects of living in space for both short and long durations of time.”. Seen the references for this overview: the papers are mainly, but not exclusively, from USA research groups. It would enrich and improve the value of the work when some more works from Russian, European, Japanese and Canadian groups would be included in order to provide a broader view on various subjects.
Response: Thank you very much for suggestions. We have added some works from other countries, however, we selected mainly landmark studies and literature in English language. Please feel free to provide important articles in your opinions and we will add accordingly.
L76: “…defined as microgravity exposure for up to 30 days,..” Where does this definition come from? Please include reference and / or clarify.
Response: Thank you very much for suggestions. This is an internal NASA protocol for short duration spaceflight meaning exposure up to 30 days.
L96: please update this “ENREF 16” reference.
Response: Thank you very much for suggestions. We believe this was an error when journal converted references into Cells format.
L101: A reference is made to Box 1: There is not Box 1 in the manuscript. Please clarify.
Response: Thank you very much for suggestions. In fact, Box 1 is Table 1. We have deleted Box 1.
Table 1 but also in other locations it is stated that countermeasures are lacking for quite some pathologies. However, for quite some time already Artificial Gravity (AG) has been postulated as adequate multi-system countermeasure especially the chronic exposure in a large radius systems. The main argument people use / think why such a countermeasure is not feasible is the huge increase in costs but there are various studies that show the opposite and also the recent decrease in launch cost makes the budget issue nearly irrelevant especially when a huge effort is paid to counteract the lack of gravity. (see e.g. Clément G, Bukley A. Springer Science & Business Media, 2007, / Clement G R, Bukley A P, Paloski W H. Artificial gravity as a countermeasure for mitigating physiological deconditioning during long-duration space missions. Front Syst Neurosci, 2015, 9: 92 / Turner A E. Orbit Dynamics and Habitability Considerations for a Space Hotel with Artificial Gravity. AIAA SPACE 2014 Conference and Exposition, 2014. 4403 / Paloski W H, Charles J B. 2014 International Workshop on Research and Operational Considerations for Artificial Gravity Countermeasures. NASA/TM-2014-217394. 2014 etc. This review should address the use of AG especially long-radius chronic AG is a possible (if not the only/best) multi-system countermeasure. Most countermeasures are directed towards cardiovascular / musculoskeletal pathologies but there is little to non against issues like immune and sleep deprivation, SANS, skin etc. etc.
Response: Thank you very much for suggestions. We have added Artificial Gravity in the discussion as suggested.
Table 1: … Psychologic distress update the : “ENREF_229”
Response: Thank you very much for suggestions. We believe this was an error when journal converted references into Cells format.
Table 2 mentions the application of Edge TPU. This is some circuitry / IoT platform that provides inferencing for low-power device. Such systems could be used in flight but for this manuscript specifically mentioned the Google Edge TPU. Why? I was wandering if it needs to be specifically a Google device? There are other systems that have the same functionality e.g. NVIDIA, Intel etc.? Please clarify and if other systems have the same / similar functionality please keep the description more general and refrain from using branding names in such manuscripts if not needed.
Response: Thank you very much for suggestions. We have added other computer brands regarding this issue in the discussion as suggested.
Table 2 last like on ethics: see also the paper by Vernikos et al. on this issue (doi: 10.3389/fphys.2020.00470)
L121: “… may cause..”: I think we can safely state “..does cause..”
Response: Thank you very much for suggestions. Done.
L184: “…ionizing radiation simulator…” : please be more specific with respect to space radiation not ‘just’ ionizing radiation.
Response: Thank you very much for suggestions. We revised as suggested.
L190: Reference #50 is not about the Twin study: Please clarify / correct.
Response: Thank you very much for suggestions. We believe this was an error that comes with ENREF”. The reference doi:10.1126/science.aau8650. has been corrected.
L198: Reference #55 does not mention microgravity for possible ROS generation. Please clarify.’
Response: Thank you very much for suggestions. We deleted as suggested.
L282: Capital “G” is used while lower case ‘g’ is meant i.e. gravity. As by the MDPI journal instructions “SI Units (International System of Units) should be used” so lower case g for gravity.
Response: Thank you very much for suggestions. We revised as suggested.
L455: update the “ENREF 172” in the text.
Response: Thank you very much for suggestions. We updated as suggested.
L353: there are no joint injuries reported in ref 139. Please clarify or correct.
Response: Thank you very much for suggestions. We updated as suggested.
L356: Ref 141 is in LBP and not specifically mentioning flexion/extension exercise. Please correct / adapt
Response: Thank you very much for suggestions. We updated as suggested.
L355: Ref 140 is not microgravity but bed rest. Please correct Also ref 140 is not a specific study but a review and does not deal with muscle imbalances: please correct.
Response: Thank you very much for suggestions. We updated as suggested.
L363: The authors mention a threshold for bone via the Frost proposed “mechanostat”. I would suggest to include the notion that, apparently, despite the current countermeasure programs the issue of skeletal integrity is still not solved: see doi: 10.1002/jbmr.3188 and doi: 10.1016/s0140-6736(00)02217-0 / 10.1038/s41526-017-0013-0. Such data should be included in this kind of review.
Response: Thank you very much for suggestions. We updated as suggested.
L418: A mention is made regarding the problem of durst for future (Moon) missions. Since it is such an important issue maybe good to include a reference on the subject.
Response: Thank you very much for suggestions. We revised as suggested.
L 577: In conclusion: “..NASA and a host of commercial space companies..”: What about other entities? Russia, Europe, Canada, Japan, China etc.? Please include / clarify.
Response: Thank you very much for suggestions. We updated as suggested.
Reference list:
There are quite some references referring to NASA documents and providing a URL. Already some of the URLs are not valid anymore. In general please use peer reviewed papers for reference and if NASA or documents from other entities cannot be avoided please include more information: title/doc number, date etc. (if available)
Also check formats: some refs are in capitals e.g. #11, #118, #262, or #273
Response: Thank you very much for suggestions. We believe this was an error that comes with formatting “ENREF”. We will double check all references in the final version of the manuscript before the proofread.
Please improve ref 96
Response: Thank you very much for suggestions. We updated as suggested.
Please clarify abbreviations: DVT, ASIC, TB, PPD, PSA, DXA, T4, TSH, GDR, VZV and maybe some others.
Response: Thank you very much for suggestions. We updated as suggested.
Supplementary info:
For Table 2: 2nd column (Inflight (short duration)) could be deleted: there are no entries.
Response: Thank you very much for suggestions. We deleted as suggested.
For Table 2: “Inflight (long duration)” What is long duration Please identify time. Also for post flight.
Response: Thank you very much for suggestions. We updated as suggested.
“Supplementary Table 3(Box 1)”: What does the ‘Box 1” mean? Please clarify / correct.
Response: Thank you very much for suggestions. We updated as suggested.
References: # 21, 22, 25, 28, 29, 30 etc. Please provide more detailed information rather than just a pdf link or URL which might not be existing after some time.
Response: Thank you very much for suggestions. We believe this was an error that comes with formatting “ENREF”. We will double check all references in the final version of the manuscript before the proofread.
General:
Any literature regarding spaceflight and (articular) cartilage is missing.
Response: Thank you very much for suggestions. We updated as suggested.
Space fever is a missing subject: see e.g. doi: 10.1038/s41598-017-15560-w
Response: Thank you very much for suggestions. We updated as suggested.
Also there are quite some recent studies regarding would healing and spaceflight. The manuscript would improve if some of this research in included.
Response: Thank you very much for suggestions. We updated as suggested.

Reviewer 2 Report
The manuscript by Krittanawong and quite some other colleagues provides a wide overview of the effects of spaceflight on human health. List of co-authors is a plethora of space-flight related experts wide a wide experience in the field.
The title does not reflect the content of the manuscript with respect to the specific attention to the subject of AI.
L66: “Here, we review the available medical literature on human experiments conducted during space travel and summarize our current knowledge on the effects of living in space for both short and long durations of time.”. Seen the references for this overview: the papers are mainly, but not exclusively, from USA research groups. It would enrich and improve the value of the work when some more works from Russian, European, Japanese and Canadian groups would be included in order to provide a broader view on various subjects.
L76: “…defined as microgravity exposure for up to 30 days,..” Where does this definition come from? Please include reference and / or clarify.
L96: please update this “ENREF 16” reference.
L101: A reference is made to Box 1: There is not Box 1 in the manuscript. Please clarify.
Table 1 but also in other locations it is stated that countermeasures are lacking for quite some pathologies. However, for quite some time already Artificial Gravity (AG) has been postulated as adequate multi-system countermeasure especially the chronic exposure in a large radius systems. The main argument people use / think why such a countermeasure is not feasible is the huge increase in costs but there are various studies that show the opposite and also the recent decrease in launch cost makes the budget issue nearly irrelevant especially when a huge effort is paid to counteract the lack of gravity. (see e.g. Clément G, Bukley A. Springer Science & Business Media, 2007, / Clement G R, Bukley A P, Paloski W H. Artificial gravity as a countermeasure for mitigating physiological deconditioning during long-duration space missions. Front Syst Neurosci, 2015, 9: 92 / Turner A E. Orbit Dynamics and Habitability Considerations for a Space Hotel with Artificial Gravity. AIAA SPACE 2014 Conference and Exposition, 2014. 4403 / Paloski W H, Charles J B. 2014 International Workshop on Research and Operational Considerations for Artificial Gravity Countermeasures. NASA/TM-2014-217394. 2014 etc. This review should address the use of AG especially long-radius chronic AG is a possible (if not the only/best) multi-system countermeasure. Most countermeasures are directed towards cardiovascular / musculoskeletal pathologies but there is little to non against issues like immune and sleep deprivation, SANS, skin etc. etc.
Table 1: … Psychologic distress update the : “ENREF_229”
Table 2 mentions the application of Edge TPU. This is some circuitry / IoT platform that provides inferencing for low-power device. Such systems could be used in flight but for this manuscript specifically mentioned the Google Edge TPU. Why? I was wandering if it needs to be specifically a Google device? There are other systems that have the same functionality e.g. NVIDIA, Intel etc.? Please clarify and if other systems have the same / similar functionality please keep the description more general and refrain from using branding names in such manuscripts if not needed.
Table 2 last like on ethics: see also the paper by Vernikos et al. on this issue (doi: 10.3389/fphys.2020.00470)
L121: “… may cause..”: I think we can safely state “..does cause..”
L184: “…ionizing radiation simulator…” : please be more specific with respect to space radiation not ‘just’ ionizing radiation.
L190: Reference #50 is not about the Twin study: Please clarify / correct.
L198: Reference #55 does not mention microgravity for possible ROS generation. Please clarify.
L282: Capital “G” is used while lower case ‘g’ is meant i.e. gravity. As by the MDPI journal instructions “SI Units (International System of Units) should be used” so lower case g for gravity.
L455: update the “ENREF 172” in the text.
L353: there are no joint injuries reported in ref 139. Please clarify or correct.
L356: Ref 141 is in LBP and not specifically mentioning flexion/extension exercise. Please correct / adapt
L355: Ref 140 is not microgravity but bed rest. Please correct Also ref 140 is not a specific study but a review and does not deal with muscle imbalances: please correct.
L363: The authors mention a threshold for bone via the Frost proposed “mechanostat”. I would suggest to include the notion that, apparently, despite the current countermeasure programs the issue of skeletal integrity is still not solved: see doi: 10.1002/jbmr.3188 and doi: 10.1016/s0140-6736(00)02217-0 / 10.1038/s41526-017-0013-0. Such data should be included in this kind of review.
L418: A mention is made regarding the problem of durst for future (Moon) missions. Since it is such an important issue maybe good to include a reference on the subject.
L 577: In conclusion: “..NASA and a host of commercial space companies..”: What about other entities? Russia, Europe, Canada, Japan, China etc.? Please include / clarify.
Reference list:
There are quite some references referring to NASA documents and providing a URL. Already some of the URLs are not valid anymore. In general please use peer reviewed papers for reference and if NASA or documents from other entities cannot be avoided please include more information: title/doc number, date etc. (if available)
Also check formats: some refs are in capitals e.g. #11, #118, #262, or #273
Please improve ref 96
Please clarify abbreviations: DVT, ASIC, TB, PPD, PSA, DXA, T4, TSH, GDR, VZV and maybe some others.
Supplementary info:
For Table 2: 2nd column (Inflight (short duration)) could be deleted: there are no entries.
For Table 2: “Inflight (long duration)” What is long duration Please identify time. Also for post flight.
“Supplementary Table 3(Box 1)”: What does the ‘Box 1” mean? Please clarify / correct.
References: # 21, 22, 25, 28, 29, 30 etc. Please provide more detailed information rather than just a pdf link or URL which might not be existing after some time.
General:
Any literature regarding spaceflight and (articular) cartilage is missing.
Space fever is a missing subject: see e.g. doi: 10.1038/s41598-017-15560-w
Also there are quite some recent studies regarding would healing and spaceflight. The manuscript would improve if some of this research in included.
Author Response

(The authors gave the same response as above.)

Reviewer 3 Report
Dear Authors.
I would like to start by congratulating you for an extensive review and manuscript as the one you have submitted for publication consideration. I have read with great interest your review, and notice that it provides a very good and comprehensive review to the field. There are perhaps three points / comments that I'd like you to consider for revision. These include;
1) As an article / review focusing on health implications in space, and counter measures, you rightly mention hematology as one of the areas important. However, you did not mention the importance of space anemia, the destructions of RBCs in vivo in astronauts in space vs. those that remain on Earth. A quick search online, will turn up a number of important publications, and cited also in major media establishments (CNN, BBC...) At least one article was written by a group in Ottawa (Guy Trudel..) I think this merits mention and entry
2) Another important point - is the use of (i) Genetic and (ii) DNA profiling - of astronauts not just before launch - but also the use of such tools during in-flight especially for long-duration flights to deeper space. The genetic makeup will invariably change to the shifting epigenetic microenvironment in-flight, future crewed-missions to deep space will have to adapt to these changes, become aware of impending red-flag situations, and whether any meaningful shift or change to ones' genetic makeup is at risk.
and so for my third point,
3) Perhaps a very small section, not more than a paragraph entry, that details about 'future' tools and methods applications for deep space missions. How these will help, assist and ensure a safer trip to deeper space missions, and more importantly, how to bring these people back to earth. Such tools and methods may initially be 'remotely' controlled or have its data sent to earth for analysis, however effort and progress should be focused on analysing data in situ, and on site during the mission itself for efficiency, and progressive purposes and slowly weaning off dependency on Earth.
Those three points, I feel - if added to your review - will strengthen the reader base, will make it more appealing, and attract a large number of Life Scientists interest' in the field of space medicine, and space bioscience.
Author Response

(The authors gave the same response as above.)
